# Fleet Active Learning:
# A Submodular Maximization Approach

**Oguzhan Akcin**[1], **Orhan Unuvar**[1], **Onat Ure**[1] **and Sandeep P. Chinchali**[1] *

**Abstract:** In multi-robot systems, robots often gather data to improve the performance of their deep neural networks (DNNs) for perception and planning. Ideally, these robots should select the most informative samples from their local data distributions by employing active learning approaches. However, when the data collection is distributed among multiple robots, redundancy becomes an issue as different robots may select similar data points. To overcome this challenge, we propose a fleet active learning (FAL) framework in which robots collectively select informative data samples to enhance their DNN models. Our framework leverages submodular maximization techniques to prioritize the selection of samples with high information gain. Through an iterative algorithm, the robots coordinate their efforts to collectively select the most valuable samples while minimizing communication between robots. We provide a theoretical analysis of the performance of our proposed framework and show that it is able to approximate the NP-hard optimal solution. We demonstrate the effectiveness of our framework through experiments on real-world perception and classification datasets, which include autonomous driving datasets such as Berkeley DeepDrive. Our results show an improvement by up to $25.0\%$ in classification accuracy, $9.2\%$ in mean average precision and $48.5\%$ in the submodular objective value compared to a completely distributed baseline.

## 1 Introduction

Consider a scenario where a fleet of autonomous vehicles (AVs) operates in various environments, ranging from urban to rural or highway settings. These AVs aim to enhance their machine learning (ML) models employed in perception, prediction, and planning by collecting data and sharing it with a central server. An ideal strategy for these AVs is to gather diverse data, enabling the trained models to generalize effectively across different environments. However, due to limitations such as bandwidth, computational resources, and storage capacity, the AVs can only transmit a limited amount of data to the central server. Notably, one AV can generate a data stream of more than 20-30 Gigabytes (GB) per second, combining video and LiDAR data [1]. In comparison, a standard 5G wireless network offers a bandwidth of only 10 Gbps, which needs to be shared among multiple users [2]. Consequently, it becomes crucial for the AVs to collaborate in selecting the most informative data points that optimize the performance of the trained models.

To illustrate, suppose one AV collects data from a congested urban area during peak hours while another AV captures data from a sparsely populated rural region. Instead of duplicating efforts in either environment, the AVs should collaborate to ensure data diversity in the collected images. By strategically selecting complementary images, the AVs contribute to the creation of a diverse dataset that enhances the robustness of the trained models across various driving scenarios.

In our formulation (Fig.1), AVs communicate with each other using minimal information and exclusively have access to their observed data, ensuring the AVs maintain data privacy and only share selected information. Moreover, we acknowledge that the collected AV data is not independent and identically distributed (i.i.d.), and the incremental value of selecting a particular data point relies on the data points already chosen. Additionally, we do not assume the presence of a perfect labeler capable of labeling data points on the fly; instead, we leverage the outputs of imperfect ML models.

---

*[1] Department of Electrical and Computer Engineering (ECE), The University of Texas at Austin, Austin, TX {oguzhanakcin,orhan.unuvar}@utexas.edu, {ureonat,sandeepc}@utexas.edu

7th Conference on Robot Learning (CoRL 2023), Atlanta, USA.

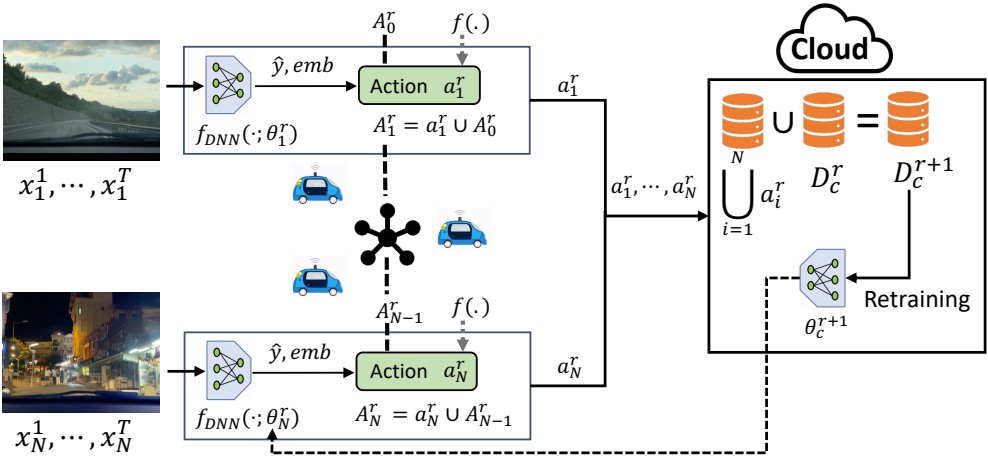

Figure 1: **Fleet Active Learning Framework:** Each robot $i$ observes a stream of data points $x_i^1, \ldots, x_i^T$ and processes them using its neural network $f_{\text{DNN}}(\cdot; \theta_i^r)$ to obtain predictions $\hat{y}$ and embeddings $emb$. Then, robots utilize these embeddings and predictions to select an action $a_i^r$ to sample data points, maximizing a submodular function $f$ while considering the previous robots' actions $A_{i-1}^r$. The aggregated action $A_i^r = a_i^r \cup A_{i-1}^r$ is passed to the next robot, repeating the process until all robots have taken an action. At the end of each round, the actions are shared with the cloud, which labels the newly acquired data points and updates the training dataset $\mathcal{D}_c^r$ with newly acquired data points $\bigcup_{i=1}^N a_i^r$, resulting in the creation of a new dataset $\mathcal{D}_c^{r+1}$. The model is retrained with the new dataset, yielding $\theta_c^{r+1}$. Finally, the cloud shares the updated model weights with all robots, and each robot updates its model parameters accordingly.

To create a framework that can be applied to a variety of data collection problems, we formulated this problem as a distributed submodular maximization problem. The reasoning behind this comes from the fact that the gain in collecting data points decreases over time, and the value of the data points is not independent but rather depends on the previously selected data points. The submodularity property captures the diminishing return in collecting data points and the dependence of the value of the data points on the previously selected data points. The main contributions of our work can be summarized as follows:

1. We present a framework to scale active learning in a multi-robot setting by formulating the data collection problem in a networked AV system as a distributed submodular maximization problem.

2. We propose an interactive algorithm that iteratively updates the actions of the AVs and provide a rigorous proof that it achieves a solution within the $1/2$ optimality bound.

3. We empirically show that our interactive algorithm performs similarly to a centralized algorithm while outperforming a fully decentralized algorithm. Our algorithm shows an improvement of up to $48.5\%$ in the submodular objective, $25.0\%$ in classification accuracy, and $9.2\%$ in mean average precision (mAP) compared to the fully decentralized approach in real-world datasets including the Berkeley DeepDrive autonomous driving dataset [3].

We also utilize embeddings generated by vision and language models such as CLIP [4], which are trained on large-scale datasets to further enhance the data point selection process. These embeddings aid in selecting data points to effectively cover the entire data distribution.

## 2 Related Work

Data collection is a well-studied problem in the robotics and machine learning literature, closely related to active learning [5–9] and cloud robotics [10–20]. However, existing approaches typically involve robots sharing *all* their data without considering other robots' input during the data collection process. The closest work to ours is [19], which addresses data collection in a multi-robot setting with the interaction between robots. However, their focus is on minimizing a convex loss function on dataset statistics, with an equal value assigned to each class point. In contrast, our framework

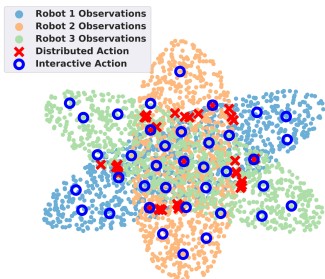

Figure 2: **Enhancing Information Coverage through Cooperation:** We illustrate a toy example involving 3 robots observing data mapped into a 2D space. Each point represents an embedding of a data point. The objective is to maximize the coverage of the observed data points through selected points. In the distributed setting (red), each robot selects a subset of data points independently, leading to overlaps and redundancy. However, when the robots cooperate using an iterative policy (blue), they consider previous actions, resulting in more diverse data collection that covers a larger portion of the data points.

provides a more general approach, considering cases where data points are unequally valued and utilizing submodular maximization for selecting the most informative samples.

Another related topic is federated learning (FL) [21–29]. FL aims to train a model using data distributed among multiple robots, where robots train the model locally and share parameters with each other. However, our problem setting fundamentally differs as the robots lack access to perfectly labeled data and need to transmit their data to the cloud to obtain labels. Additionally, the models we consider are considerably large, making local training infeasible.

Our work predominantly falls into the category of pool-based active learning [7, 30], where the objective is to select the most informative data points from an unlabeled pool. Various methods have been proposed for selecting these data points in general machine learning settings [31–37] and robotics [38–41], each capturing informativeness in different ways and also requiring access to all unlabeled data points. In our case, however, we consider a multi-robot setting where robots only have access to their local data distribution. Our work extends these methods naturally to be applicable in a multi-robot context.

The problem of active learning is closely related to submodular maximization [42, 43], as submodular functions capture the diminishing returns property and the marginal gain of selecting a data point depends on the previously selected points [34, 43–46]. Numerous works address submodular maximization in both decentralized [42, 47–50] and centralized settings [43, 51–57]. Specifically, the works such as [49, 50] investigate a similar problem to ours. However, they restrict robots to select actions with a single element, whereas our work broadens this to accommodate actions with multiple elements. Our extension is essential since AVs upload a batch of sampled data to labelers over bandwidth-limited networks. Furthermore, the problems in [49, 50] address classical submodular optimization, like set cover and sensor placement. Conversely, our work focuses on active learning for ML models and distributed data collection.

## 3   Problem Formulation

We consider a practical problem that involves data collection from a distributed fleet of robots for training a robust machine learning model in the cloud. Our goal is to select the most informative samples from each robot in the fleet, such that when these data points are added to the training dataset, the accuracy of the trained model is maximized. This problem formulation can be seen as a generalized version of the active learning problem in the context of a distributed fleet of robots.

**Robot Perception Model:**   We assume a fleet of $N_{\mathrm{robot}}$ robots operating in diverse environments, each performing a computer vision task such as classification or object detection. Robot $i$ observes a set of data points $\{x_i^1, \ldots, x_i^T\} = X_i^r$, which represents the accumulated data available for sampling in round $r$. A round represents a data collection period, like a day, during which robots operate and gather data. In each round, the robots sample $N^{\mathrm{cache}}$ data points from their accumulated data $X_i^r$ and send them to the cloud to retrain an ML model. The perception model used by robot $i$ is a DNN represented as $f_{\mathrm{DNN}}(x; \theta_i^r) = \hat{y}, emb$, where $\theta_i^r$ denotes the parameters of the perception model. The perception model provides predictions $\hat{y}$ and embeddings $emb$ for the input data $x$.

Embeddings are crucial in our approach as they capture the similarity between data points. Unlike raw data, which is high-dimensional and ineffective at summarizing information, embeddings provide a more efficient and accurate representation. By leveraging embeddings generated by foundation models like CLIP [4] or other active learning methods [32–34, 58], we enhance our ability to capture data similarity effectively.

**Assumption 1** (Robots' Accumulated Data Points are Disjoint). *Each robot $i$ stores its observed data $X_i^r$ separately, resulting in disjoint sets of accumulated data points for selection in each round $r$. That is, $X_i^r \cap X_j^r = \emptyset$ for all $i \neq j$.*

This assumption reflects the fact that each robot maintains its own storage of observed data points. As a result, the sets of accumulated data available for selection in each round are disjoint. Then for each round $r$, we can define the set of all observed images as $X^r = \bigcup_{i=1}^{N_{\text{robot}}} X_i^r$.

**Robotic Fleet:** The fleet of $N_{\text{robot}}$ robots operate in diverse environments, observing different data distributions $X_i^r$ in each round $r$. In each round, each robot $i$ selects at most $N_i^{\text{cache}}$ data points from its observed images $X_i^r$ and sends them to the cloud for retraining the ML model. The size of the selected samples $N_i^{\text{cache}}$ is determined based on factors such as communication bandwidth and labeling budget. The samples chosen by robot $i$ in round $r$ are denoted as action $a_i^r$, where $a_i^r \subset X_i^r$ and $|a_i^r| \leq N_i^{\text{cache}}$, indicating that the number of selected samples is smaller than or equal to the cache size $N_i^{\text{cache}}$. Additionally, we assume that the robotic fleet is connected via a wireless network, enabling communication among the robots.

**Definition 1** (Robots' Feasible Action Space). *In round $r$, each robot $i$ can select at most $N_i^{\text{cache}}$ samples from the observed datapoints $X_i^r$ such that:*

$$\mathcal{A}_i^r = \{a_i^r \subset X_i^r : |a_i^r| \leq N_i^{\text{cache}}\}, \quad \mathcal{A}^r = \{\bigcup_{i=1}^{N_{\text{robot}}} a_i^r : a_i^r \in \mathcal{A}_i^r, \forall i = 1, \ldots, N_{\text{robot}}\}.$$

*Here, $\mathcal{A}_i^r$ represents the set of all feasible actions for robot $i$, and $\mathcal{A}^r$ represents the combined feasible actions of all robots.*

**Data Collection Problem:** The goal of the data collection process is to select the most informative samples from each robot in the fleet, such that when these samples are added to the training set, the accuracy of the trained model is maximized. However, there is no proven function that reliably represents dataset quality, as it depends on various factors such as data diversity and image quality [59]. Instead, heuristic methods are used to measure dataset quality. To provide a general framework for the data collection problem, we will assume two properties about the dataset quality.

First, we will assume that the quality of the dataset always increases with the addition of new samples [43–46, 60]. This is a reasonable assumption since, unless the training data points are intentionally manipulated, adding more data points improves the coverage of the training set of the real data distribution. Thus, we can safely make the following assumption:

**Assumption 2** (Dataset Quality Function is Monotone). *The dataset quality function $f : 2^X \to \mathbb{R}$ is a monotone function, meaning that for all $A \subseteq B \subseteq X$, $f(A) \leq f(B)$.*

Secondly, we assume that the marginal gain of adding a new sample to the training dataset exhibits a diminishing returns [43–46]. This is also a reasonable assumption because if the training dataset already covers a significant portion of the real data distribution, adding more samples may not increase the accuracy of the trained model significantly. We verified this on four real-world datasets, confirming a diminishing returns relationship between test accuracy and the percentage of training data (App. A.6). This property, known as submodularity, is exhibited by many problems, such as set cover, facility location, and sensor placement [51]. Formally, we state the following assumption:

**Assumption 3** (Dataset Quality Function is Submodular). *The dataset quality function $f : 2^X \to \mathbb{R}$ is submodular, i.e., for all $A \subseteq B \subseteq X$ and $x \in X \setminus B$ it holds that $f(A \cup \{x\}) - f(A) \geq f(B \cup \{x\}) - f(B)$.*

Finally, we define the data collection problem: given a fleet of $N_{\text{robot}}$ robots sharing a common dataset $\mathcal{D}_c^r$ in the cloud, the objective is to select a maximum of $N_i^{\text{cache}}$ samples from each robot $i$, maximizing the quality of the dataset. Formally, we can define the data collection problem as:

**Problem 1** (The Data Collection Problem).

$$\max_{a_1^r, \ldots, a_{N_{\text{robot}}}^r} f(\mathcal{D}_c^r \cup \bigcup_{i=1}^{N_{\text{robot}}} a_i^r) \tag{1}$$

$$\text{subject to:} \quad a_i^r \subseteq X_i^r \qquad \forall i = 1, \ldots, N_{\text{robot}}$$

$$|a_i^r| \leq N_i^{\text{cache}} \quad \forall i = 1, \ldots, N_{\text{robot}}.$$

This problem is a combinatorial optimization problem and, without any further assumptions, solving for the optimal solution is NP-hard [61]. Thus, we use algorithms that provide approximation guarantees to the optimal solution. For such problems, the greedy algorithm is shown to provide a good approximation to the optimal solution [53].

There are various submodular functions that can be used; such examples are facility location, mutual information, or the set cover problem [51]. In our experiments, we used a facility location function, which assesses how well a subset $A$ represents the entire dataset $X$ [62]. It accomplishes this by calculating the sum of similarities $M_{x,a}$ between each element $x \in X$ and its closest element $a \in A$. The facility location function is known to be submodular for non-negative similarity values $M_{x,a} \geq 0$ [63]. Fig. 2 shows an example of how our facility location submodular objective encourages covering the data distribution. In formal terms, the facility location function is defined as follows:

$$f(A) = \sum_{x \in X^r} \max_{a \in A} M_{x,a}. \tag{2}$$

**Model Retraining and Updating Weights:** After each data collection round $r$, the cloud dataset $\mathcal{D}_c^r$ is expanded with selected samples, and the perception model is retrained on the cloud using the updated dataset $\mathcal{D}_c^{r+1} = \mathcal{D}_c^r \cup \bigcup_{i=1}^{N_{\text{robot}}} a_i^r$ to obtain new model weights $\theta_c^{r+1}$. These weights are then shared with the robots. The robots can simply use the updated cloud model $\theta_c^{r+1}$ or optionally fine-tune it on their local datasets to yield models $\theta_i^{r+1}$.

# 4 Baselines

**Centralized Action Policy:** We now introduce a centralized action policy, named CENTRALIZED, described in Alg. 1. In this policy, a centralized cloud has access to all the data points observed by the robots $X^r$ (line 4). Starting from an empty solution set (line 3), the central cloud iteratively evaluates all combined data points $X^r$, selects the data point within the feasible action space that maximizes the submodular function $f$ (line 6) and adds it to the action (line 7). This process continues until no more points can be added to the action (lines 5 - 8). The action of the CENTRALIZED policy, denoted as $a^C$, is then appended to the cloud dataset $\mathcal{D}_c^r$ (line 9), and this algorithm is repeated for each round $r$.

This algorithm has been proven to provide $1/2$ optimality bound to the optimal solution [53]. Additionally, randomized approximations exist that can achieve a $1 - 1/e$ optimality bound [52]. We have chosen this version of the centralized policy because of its simplicity and generality. Since solving the combinatorial optimization problem is NP-hard, we will use this centralized policy as a target benchmark to compare the performance of our framework.

---

1 **Input:** $\mathcal{D}_c^r, X_i^r, f$ ;
2 **Output:** $\mathcal{D}_c^{r+1}$ ;
3 Initialize $a^C = \emptyset$ ;
4 $X^r = \bigcup_{i=1}^{N_{\text{robot}}} X_i^r$ ;
5 **for** $j = 1$ *to* $\sum_{i=1}^{N_{\text{robot}}} N_i^{\text{cache}}$ **do**
6     $x_j = \underset{x : a^C \cup \{x\} \in \mathcal{A}^r}{\operatorname{argmax}} f(\mathcal{D}_c^r \cup a^C \cup \{x\})$ ;
7     $a^C = a^C \cup \{x_j\}$ ;
8 **end**
9 $\mathcal{D}_c^{r+1} = \mathcal{D}_c^r \cup a^C$ ;
10 **return** $\mathcal{D}_c^{r+1}$ ;

**Algorithm 1:** CENTRALIZED Policy

---

1 **Input:** $\mathcal{D}_c^r, X_i^r, f$ ;
2 **Output:** $\mathcal{D}_c^{r+1}$ ;
3 **for** $i = 1$ *to* $N_{\text{robot}}$ **do**
4     Initialize $a_i^D = \emptyset$ ;
5     **for** $j = 1$ *to* $N_i^{\text{cache}}$ **do**
6        $x_j = \underset{x : a_i^D \cup \{x\} \in \mathcal{A}_i^r}{\operatorname{argmax}} f(\mathcal{D}_c^r \cup a_i^D \cup \{x\})$ ;
7        $a_i^D = a_i^D \cup \{x_j\}$ ;
8     **end**
9 **end**
10 $\mathcal{D}_c^{r+1} = \mathcal{D}_c^r \cup \bigcup_{i=1}^{N_{\text{robot}}} a_i^D$ ;
11 **return** $\mathcal{D}_c^{r+1}$ ;

**Algorithm 2:** DISTRIBUTED Policy

---

**Distributed Action Policy:** In another scenario, we can solve this problem without considering any communication between robots, referred to as DISTRIBUTED . This policy is described in Alg. 2. In the DISTRIBUTED policy, each robot independently runs the greedy selection in its observed images $X_i^r$ in parallel (lines 3 - 9). Starting with an empty solution (line 4), each robot adds data points that maximize the submodular function $f$ from its observed data points (line 5). At the end of round $r$, the selected data points $a_i^D$ are shared with the cloud, which adds them to the cloud dataset

$\mathcal{D}_c^r$ (line 11). Although the DISTRIBUTED policy can be run in parallel and preserves the distributed nature of the problem, it does not consider other robots' actions, potentially resulting in repeatedly selecting similar elements. Consequently, the solutions obtained through the DISTRIBUTED policy can be an $\Omega(N_{\text{robot}})$ factor worse than the centralized solution [42].

These algorithms offer different approaches for sample selection from distributed robots, with the CENTRALIZED policy utilizing global information and the DISTRIBUTED policy relying on local observations of each robot. The CENTRALIZED policy needs $O(\sum_{i=1}^{N_{\text{robot}}} N_i^{\text{cache}})$ iterations over all robots and their samples, which may be impractical for a large number of robots. On the other hand, the DISTRIBUTED policy can be implemented in parallel, making it more scalable. Our method balances these two policies using a distributed policy with an iterative component.

## 5 An Interactive Approach to Fleet Active Learning

This section introduces our proposed policy, called INTERACTIVE , for the data collection problem. The INTERACTIVE policy begins with an empty solution (line 1) and sequentially determines the actions of the robots (lines 4 - 7). Each robot $i$ takes into account the previous actions of the robots 1 to $i-1$, denoted as $A_{i-1}^I$, and runs the greedy algorithm given in Alg.3 to determine its action $a_i^I$ and includes it in the previous actions $A_i^I$ (line 6). This process is repeated until all robots have determined their actions. Finally, the actions of all the robots are combined (line 8), resulting in the combined action $a^I$, and the cloud dataset is updated (line 9).

---

**1 Input:** $\mathcal{D}_c^r, X_i^r, f , A_{i-1}^I$ ;
**2 Output:** $a_i^I$ ;
**3** $a_i^I = \emptyset$ ;
**4 for** $j = 1$ *to* $N_i^{\text{cache}}$ **do**
**5** $\quad x_j = \underset{x:a_i^I \cup \{x\} \in \mathcal{A}_i^r}{\operatorname{argmax}} f(\mathcal{D}_c^r \cup A_{i-1}^I \cup a_i^I \cup \{x\})$ ;
**6** $\quad a_i^I = a_i^I \cup \{x_j\}$ ;
**7 end**
**8 return** $a_i^I$ ;

**Algorithm 3:** Action per Robot

**1 Input:** $\mathcal{D}_c^r, X_i^r, f$ ;
**2 Output:** $\mathcal{D}_c^{r+1}$ ;
**3** $A_0^I = \emptyset$ ;
**4 for** $i = 1$ *to* $N_{\text{robot}}$ **do**
**5** $\quad$ Get action $a_i^I$ using algorithm 3 ;
**6** $\quad A_i^I = A_{i-1}^I \cup a_i^I$ ;
**7 end**
**8** $a^I = \bigcup_{i=1}^{N_{\text{robot}}} a_i^I$ ;
**9** $\mathcal{D}_c^{r+1} = \mathcal{D}_c^r \cup a^I$ ;
**10 return** $\mathcal{D}_c^{r+1}$ ;

**Algorithm 4:** INTERACTIVE Policy

---

Our proposed INTERACTIVE policy represents an improvement over both the CENTRALIZED and DISTRIBUTED policies. It achieves a solution with the same optimality bound as the CENTRALIZED policy while allowing for distributed execution. Moreover, the selection of points occurs only once between robots, leading to a reduced number of messages passings between robots. To minimize data exchange, we can share the embeddings of the points instead of the actual points themselves [15]. This significantly reduces the amount of data transmitted. Furthermore, for specific functions like the facility location function, additional optimizations can be employed. For example, rather than sharing the points or embeddings, robots can update and share the maximum value of the similarity metric $M_{x,a}$. We also theoretically show that the given algorithm will achieve a solution that is at least $1/2$ good as the optimal solution for submodular monotone objective functions.

**Theorem 1** (Optimality of INTERACTIVE Policy). *The* INTERACTIVE *policy described in Alg. 4 achieves a solution that is at least $1/2$ as good as the optimal solution for submodular monotone objective functions, that is:*

$$f_{\mathcal{D}_c^r}(a^I) \geq \frac{1}{2} f_{\mathcal{D}_c^r}(a^{\text{OPT}}). \tag{3}$$

*Here we denote the improvement on the objective function as $f_{\mathcal{D}_c^r}(x) = f(\mathcal{D}_c^r \cup x) - f(\mathcal{D}_c^r)$. $a^{\text{OPT}}$ is the optimal solution to the optimization problem given in Eq. 1 and $a^I$ is the solution obtained by the* INTERACTIVE *policy.*

The proof of the theorem is given in Appendix A.2. As demonstrated in the theorem, the optimality bound of the INTERACTIVE policy is independent of the number or ordering of the robots. Furthermore, the number of messages exchanged between robots is $O(N_{\text{robot}})$. This subtle, yet significant, difference between the CENTRALIZED policy and the INTERACTIVE policy enables the INTERACTIVE policy to be scalable to a large number of robots. Moreover, our INTERACTIVE policy is

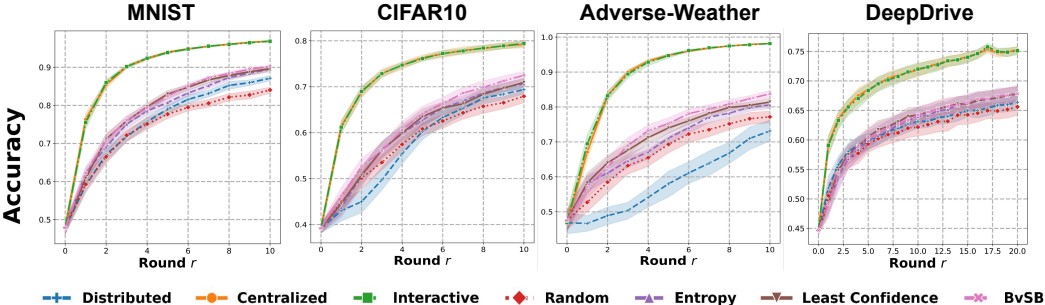

Figure 3: **Our INTERACTIVE policy achieves similar performance with the CENTRALIZED policy and outperforms other benchmarks.** This figure presents the accuracy of the retrained neural networks using the dataset $\mathcal{D}_c^r$ in each round $r$, with each column representing a different dataset. It can be clearly seen that INTERACTIVE and CENTRALIZED policies consistently outperform other benchmark policies across all datasets, underscoring the practical advantages of our INTERACTIVE policy.

computationally efficient compared to the CENTRALIZED policy and shares the same complexity as the DISTRIBUTED policy. In terms of function evaluations, both INTERACTIVE and DISTRIBUTED policies have a computational complexity of $O(\sum_{i=1}^{N_{\text{robot}}} |X_i^r| \times N_i^{\text{cache}})$, while the CENTRALIZED policy has a complexity of $O(\sum_{i=1}^{N_{\text{robot}}} |X_i^r| \times \sum_{i=1}^{N_{\text{robot}}} N_i^{\text{cache}})$. Further analysis and a numerical example are provided in the Appendix.

## 6 Experiments

We evaluate our method by comparing it against several benchmarks using four diverse datasets. The first two datasets, MNIST [64] and CIFAR-10 [65], serve as foundational benchmarks, while the Adverse-Weather dataset [66] and the DeepDrive dataset [3] provide realistic scenarios with data captured by autonomous vehicles. In our simulations, we simulate the data observation processes under heterogeneous environments and diverse data distributions. For classification tasks, we employ state-of-the-art vision models such as the Vision Transformer [67] and ResNet [68], while for object detection tasks, we utilize YOLOv8 [69]. Detailed information regarding our experimental setup and the models is given in the Appendix.

In addition to the baselines described in Section 4, we provide a comparison between our method and classical active learning approaches. These approaches include: *i*) **Random:** Selecting data points uniformly at random from the observations. *ii*) **Least Confidence:** Choosing samples with the smallest predicted class probabilities [70]. *iii*) **Best versus Second Best (BvSB):** Picking samples with the smallest difference between the two most probable classes [71]. *iv*) **Entropy:** Selecting samples with the highest entropy in predictive class probabilities [70]. The latter three methods fall within the category of uncertainty-based active learning techniques [7].

**Comparison Metrics:** For classification tasks, we compare the accuracy of the trained model on newly acquired datasets $\mathcal{D}_c^{r+1}$ using the held-out test dataset. We compare the mean Average Precision (mAP) for object detection tasks among different data sampling policies. We also compare the submodular objective value of selected data points in each round. All experiments are repeated with 25 different seeds and the results are averaged.

| Policy | mAP |
|---|---|
| INITIAL | 33.2 |
| DISTRIBUTED | 37.2 |
| CENTRALIZED | 46.5 |
| INTERACTIVE | 46.4 |

Table 1: Results of DeepDrive object detection.

**Submodular Objective Function:** We use the facility location function, given in Eq. 2, as the objective function. To measure the similarity between points $x$ and $a$, we employ the similarity metric $M_{x,a} = \frac{1}{1+\beta d(x,a)}$. Here, the distance $d(x,a)$ is computed as the $L_2$ distance between the embeddings of the data points, given by $\|emb_x - emb_a\|_2$. The embeddings are obtained by the models $f_{\text{DNN}}$ and we set the hyperparameter $\beta = 0.01$ to control the significance of the distance metric. In our simulations, we employ the lazy greedy algorithm [54], a variant of the greedy algorithm that achieves the same solution while being more efficient in practice.

**Results:** The experimental results (Fig. 3) show that our INTERACTIVE policy performs as well as the CENTRALIZED policy, as we have shown in the Theorem 1. Moreover, our INTERACTIVE policy consistently outperforms the DISTRIBUTED policy in all simulations.

***What is the accuracy of the vision model after each round?***

In Fig.3, we show the perception model's accuracy after each retraining round on the dataset $\mathcal{D}_c^r$. Clearly, our INTERACTIVE policy consistently achieves higher accuracies compared to the DISTRIBUTED policy, outperforming it by $9.7\%, 10.1\%, 25.0\%, 8.8\%$ and achieves similar accuracy values to the CENTRALIZED case. Furthermore, our INTERACTIVE policy exhibits significant accuracy improvements, with gains of up to $21.0\%$, $17.6\%$, $16.7\%$, and $14.3\%$ in comparison to the Random, Entropy, Least Confidence, and BvSB active learning methods, respectively.

***What is the object detection performance?***

In Table 1, we present the mean average precision (mAP) of the initial DNN, denoted as INITIAL , and at the end of all data collection rounds for each policy. Our INTERACTIVE policy and CENTRALIZED policy show significantly better performance compared to the DISTRIBUTED policy, surpassing it by $9.2\%$.

**Limitations:** Our formulation, while effective, has several assumptions that limit its generality. Firstly, our formulation assumes that the data collection rounds are synchronous among all robots and that robots have communication capabilities with each other. Finally, we assume that the robots are trying to maximize the objective function cooperatively, whereas, in a real-world setting, the robots may need to be incentivized based on their contributions.

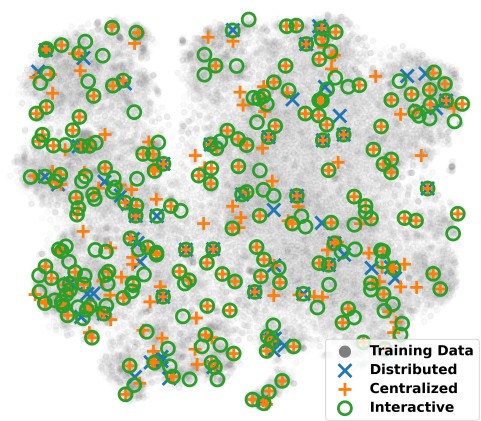

Figure 4: **Visualization of Policy Coverage in Feature Space:** We employed t-SNE [62] to project the embeddings of data points into a 2D space, facilitating the comparison of coverage across different policies. Data points are color-coded according to the policy that selected them, with INTERACTIVE (green), CENTRALIZED (orange), and DISTRIBUTED (blue) policies represented. The visualization demonstrates that our proposed INTERACTIVE method performs on par with the CENTRALIZED baseline and achieves superior coverage of the training data (gray) in comparison to the DISTRIBUTED policy. Although the DISTRIBUTED policy might seem to have fewer points, this can be attributed to the overlapping and repetition of similar data points that are closely mapped together in the visualization.

# 7    Conclusion and Future Work

We propose a framework to scale active learning algorithms to the multi-robot setting utilizing submodular maximization. Our framework enables robots to determine their actions sequentially, taking into account the actions of the previous robots. We show that when the objective function is submodular and monotone, the proposed framework achieves $1/2$ optimality bound to the optimal solution, which is NP-hard to compute. Through experiments on real-world datasets, we confirm that such an iterative algorithm will result in increased accuracy and objective function when compared to the distributed setting. Our work is especially useful when the robots are operating in a resource-constrained environment, where data sharing is costly, and collective actions need to be optimized to reduce redundancy. We believe that our work is a step towards the development of a general framework for active learning in multi-robot systems.

Future research directions include incorporating more realistic constraints. This involves incorporating asynchronous data collection rounds, where the robots can collect data at different times, and incorporating privacy-preserving mechanisms to ensure confidentiality. Furthermore, we aim to extend our formulation where robots collect data for multiple tasks, each with varying value and importance.

## Acknowledgements

This material is based upon work supported in part by Lockheed Martin Corporation and a gift from Cisco Systems, Inc. Additionally, this material received support from the National Science Foundation under grant no. 2148186 and is further supported by funding provided by federal agencies and industry partners as specified in the Resilient & Intelligent NextG Systems (RINGS) program. This article solely reflects the opinions and conclusions of its authors and does not represent the views of any sponsor.

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

# A Appendix

**Code Availability:** The code and related materials can be found in the following code repository:

https://github.com/UTAustin-SwarmLab/Fleet-Active-Learning.git

The organization of the appendix is as follows:

## A.1 Preliminaries

Here, we provide basic definitions of submodular functions, monotone functions, and matroids that we use in our proof.

**Definition 2** (Submodular Function). *A set function $f : 2^X \to \mathbb{R}$ is submodular if for all $A \subseteq B \subseteq X$ and $x \in X \setminus B$, we have*

$$f(A \cup \{x\}) - f(A) \geq f(B \cup \{x\}) - f(B).$$

**Definition 3** (Monotone Function). *A set function $f : 2^X \to \mathbb{R}$ is monotone if for all $A \subseteq B \subseteq X$, we have*

$$f(A) \leq f(B).$$

Based on the definitions of submodular and monotone functions, we can directly write the following:

**Corollary 1.** *Let $f$ be a monotone submodular function. Then, $f$ is also subadditive, and we have that:*

$$\forall A \subset X, \; B \subset X : \quad f(B) \leq f(A) + \sum_{x \in B \setminus A} f_A(x).$$

Here, we denote the marginal gain of adding an element to set $A$ as $f_A(x) = f(A \cup \{x\}) - f(A)$, which is also monotone and submodular.

*Proof.* This is a standard result in submodularity. See Corollary 5 in [72]. □

**Definition 4** (Matroid). *A matroid is a pair $M = (E, I)$ where $E$ is a finite set (called the ground set) and $I$ is a nonempty set of subsets of $E$ (called the set of independent sets) with the following conditions:*

1. $\forall B \in I : \quad A \subset B \Rightarrow A \in I.$

2. $\forall A, B \in I : \quad |A| < |B| \Rightarrow \exists x \in B \setminus A : A \cup \{x\} \in I.$

**Definition 5** (Basis of a Matroid). *A basis of a matroid is an independent set of the matroid which is not contained in any other independent set.*

**Corollary 2.** *If $B_1$ and $B_2$ are two bases of a matroid $M$, then there exists a bijection $\phi : B_1 \setminus B_2 \to B_2 \setminus B_1$ such that:*

$$\forall x \in B_1 \setminus B_2 : \quad B_1 \cup \phi(\{x\}) \setminus \{x\} \in I.$$

*Proof.* This is a standard result in matroid theory. See Proposition 11 in [72]. $\qquad\square$

**Definition 6** (Partition Matroid). *A partition matroid is a matroid where the ground set $E$ is partitioned into $l$ disjoint subsets $E_1, E_2, \ldots, E_l$ and the set of independent sets is defined as follows:*

$$I = \{B \subseteq E : |B \cap E_i| \leq k_i, \forall i \in \{1, 2, \ldots, l\}\}.$$

### A.2   Proof of Optimality Bound for the Interactive Policy

First, we show that the robots' feasible action spaces $\mathcal{A}^r$ and the union set of all observed data $X^r$ form a partition matroid.

**Lemma 2** (Robots' feasible action space and set of all observed images form Partition Matroid). *The robots' feasible action space $\mathcal{A}^r$ and the union of all observed data $X^r$ form a partition matroid $M^r = (X^r, \mathcal{A}^r)$, where $X^r = \bigcup_{i=1}^{N_{\text{robot}}} X_i^r$ and $X_i^r \cap X_j^r = \emptyset \;\; \forall i \neq j$. The action space is defined as $\mathcal{A}^r = \{\bigcup_{i=1}^{N_{\text{robot}}} a_i^r : a_i^r \in \mathcal{A}_i^r \quad \forall i = 1, \ldots, N_{\text{robot}}\}$.*

*Proof.* The proof follows directly from the definition of action and observed data points. For all $a \in \mathcal{A}^r$, if $s \subset a$, then we know that for all $j$, $|s \cap X_j^r| \leq |a \cap X_j^r| \leq N_j^{\text{cache}}$ meaning $s \in \mathcal{A}^r$. And we know that for all $a, s \in \mathcal{A}^r$, if $|s| < |a|$, then there exists a subset $X_j^r \subset X^r$ such that $|s \cap X_j^r| < |a \cap X_j^r| \leq N_j^{\text{cache}}$ and $X_j^r \cap a \setminus s \neq \emptyset$. Then, for any element $x \in X_j^r \cap a \setminus s$, it holds that $s \cup \{x\} \in \mathcal{A}^r$. $\qquad\square$

Now we prove the main theorem of our paper. Our proof is similar to the proof given in [72], while our proof involves sequential optimization methods, such as the one shown in Alg. 4.

**Theorem.** *The algorithm given in Alg. 4 achieves a solution greater than $1/2$ of the optimal solution.*

*Proof.* Assume that $a^{\text{OPT}}$ is the optimal solution for the problem 1. First, we show that $a^{\text{OPT}}$ and $a^I$ are bases of the matroid $(X^r, \mathcal{A}^r)$. Since we assume that the objective function $f$ is monotone (Assmp. 2), it is trivial to see that $|a^{\text{OPT}} \cap X_i^r| = N_i^{\text{cache}}$ for all $i \in \{1, \ldots, N_{\text{robot}}\}$, making $a^{\text{OPT}}$ a basis of matroid $(X^r, \mathcal{A}^r)$. $a^I$ is a basis as well, since in Alg. 4 we construct it such that $|a^I \cap X_i^r| = N_i^{\text{cache}}$ for all $i$. For matroids, there exists a bijection $\phi : a^{\text{OPT}} \to a^I$, which maps the optimal solution to the solution of the INTERACTIVE policy. We can express these solution sets as follows:

$$a^{\text{OPT}} = \{x_{1,1}^{\text{OPT}}, x_{1,2}^{\text{OPT}}, \ldots, x_{N_{\text{robot}}, N_{N_{\text{robot}}}^{\text{cache}}}^{\text{OPT}}\} \quad \text{and} \quad a^I = \{x_{1,1}, x_{1,2}, \ldots, x_{N_{\text{robot}}, N_{N_{\text{robot}}}^{\text{cache}}}\}.$$

Here $x_{i,j} = \phi(x_{i,j}^{\text{OPT}})$ for all $i, j$. Let $a_{i,j}^I = \{x_{1,1}, \ldots, x_{i,j}\}$ and $a_{i,0}^I = \{x_{1,1}, \ldots, x_{i-1, N_{i-1}^{\text{cache}}}\}$ denote the sets of actions taken up to the $i$-th robot and the $j$-th cache and actions taken up to the $i$-th robot repectively. Then for $f_{\mathcal{D}_c^r}(x) = f(\mathcal{D}_c^r \cup \{x\}) - f(\mathcal{D}_c^r)$, we can write the following:

$$\begin{aligned}
f_{\mathcal{D}_c^r}(a^{\text{OPT}}) - f_{\mathcal{D}_c^r}(a^I) &\leq \sum_{i=1}^{N_{\text{robot}}} \sum_{j=1}^{N_i^{\text{cache}}} f_{\mathcal{D}_c^r \cup a^I}(x_{i,j}^{\text{OPT}}) \\
&\leq \sum_{i=1}^{N_{\text{robot}}} \sum_{j=1}^{N_i^{\text{cache}}} f_{\mathcal{D}_c^r \cup a_{i,j-1}^I}(x_{i,j}^{\text{OPT}}) \\
&\leq \sum_{i=1}^{N_{\text{robot}}} \sum_{j=1}^{N_i^{\text{cache}}} f_{\mathcal{D}_c^r \cup a_{i,j-1}^I}(x_{i,j}) \\
&= \sum_{i=1}^{N_{\text{robot}}} \sum_{j=1}^{N_i^{\text{cache}}} f_{\mathcal{D}_c^r}(a_{i,j}^I) - f_{\mathcal{D}_c^r}(a_{i,j-1}^I) = f_{\mathcal{D}_c^r}(a^I).
\end{aligned}$$

The first inequality is a result of Corollary 1, while the second inequality stems from the submodularity of the function $f_{\mathcal{D}_c^r}$. In the third inequality, we utilize the fact that in each iteration of Alg. 4, we select the element with the maximum marginal gain. Next, in the first equality, we use the fact that $a_{i,j-1}^I \cup \{x_{i,j}\} = a_{i,j}^I$. The last equality follows from the fact that for $f_{\mathcal{D}_c^r}(\emptyset) = 0$, the sum of the marginal gains equals to the value of $f_{\mathcal{D}_c^r}(a^I)$.

Therefore we have:

$$f_{\mathcal{D}_c^r}(a^I) \geq \frac{1}{2} f_{\mathcal{D}_c^r}(a^{\mathrm{OPT}}).$$

This concludes the proof, showing that the INTERACTIVE policy achieves at least half of the value of the optimal solution. □

## A.3 Experiments

To demonstrate the effectiveness of our proposed policy, we conducted simulations in scenarios involving multiple robots engaged in data collection from heterogeneous observation distributions. To create these heterogenous environments, we sampled incoming class distributions from the Dirichlet distribution, incorporating a skewness parameter denoted as $\alpha$. This way, we ensure that environments have nonidentical incoming class distributions (non-i.i.d). Then, within each environment, we simulated robots that observed the same data points.

Initially, an initial dataset denoted as $\mathcal{D}_c^0$ was chosen from the training set to train the initial model $f_{\mathrm{DNN}}(.; \theta_i^0)$. The initial dataset was generated with uniform class distribution, employing the Dirichlet distribution with a skewness parameter value of $\alpha = 5$. In each subsequent round, the vision model was retrained from the pre-trained vision model to ensure a fair evaluation of the performance using the selected training set. To prevent overfitting, when identical data points were selected from multiple devices, the redundant instances were filtered out, and only a single data point was added to the training set.

### A.3.1 Embedding Functions:

To generate embeddings for the data points, we utilized multiple vision and language models depending on the datasets. Initially, we made use of the embeddings generated by the CLIP model [4], which is trained to create outputs in the same embedding space for both language and vision model inputs. However, we observed that when the embeddings generated by the CLIP model start to perform poorly on the datasets when there is a mismatch of the targets of the CLIP model with our classification output or the images are out-of-distribution for the CLIP model. For these datasets, we instead employ the embeddings generated by BADGE [33]. BADGE embeddings essentially correspond to the gradients of the final layer of the network with respect to the input.

### A.3.2 Classification Experiments

In all classification experiments, we used the Adam optimizer with a learning rate of 0.001 with a batch size of 1000. Additionally, the learning rate scheduler is used with a decay rate of 0.99. We trained the DNNs in each round for 300 epochs. We did not apply any data augmentation to the training set. To ensure robustness, we conducted these experiments for 25 different seeds. Now, we provide additional explanations regarding the details and dataset-specific parameters used in the simulations.

**MNIST:** In our paper, we used the `MNIST` dataset to show the efficacy of our algorithm in a simple setting. The `MNIST` dataset is a collection of handwritten digits that contains 60,000 training images and 10,000 test images. Each image is a $28 \times 28$ grayscale image.

**Simulation Parameters:** In `MNIST` simulations, we used 5 heterogeneous environments, each containing 4 robots that observe identical samples, resulting in a total system of 20 robots. To create heterogenous incoming class distributions, we set the skewness parameter of the Dirichlet distribution to $\alpha = 1.3$. In each round, robots are observing 1000 data samples and collect $N^{\mathrm{cache}} = 3$ data samples from their observations. We started with an initial dataset of size 16 and collected the data for 10 rounds. The final training dataset consists of 616 data points.

**DNN and Embedding Function:** We used a simple DNN with 6 layers, with 4 convolutional layers and 2 fully connected layers. Between each convolutional layer, we used the ReLU activation function and applied dropout with a probability of 0.3. To create embeddings, we utilized BADGE [33].

**CIFAR-10:** The `CIFAR-10` dataset consists of 60,000 $32 \times 32 \times 3$ RGB images with ten different classes. The dataset is split into training and testing datasets of size 50,000 and 10,000, respectively. The classes in the dataset are truck, ship, horse, frog, dog, deer, cat, bird, automobile, and airplane.

*Simulation Parameters:* In `CIFAR-10` simulations, we used 6 heterogeneous environments, each containing 4 robots that observe identical samples, resulting in a total system of 24 robots. To create heterogenous incoming class distributions, we set the skewness parameter of the Dirichlet distribution to $\alpha = 1.6$. In each round, robots are observing 1000 data samples and collect $N^{\text{cache}} = 1$ data samples from observations. We started with an initial dataset of size 10 and collected the data for 10 rounds. The final training dataset consists of 250 data points.

*DNN and Embedding Function:* We leveraged a pre-trained ResNet-50 model [68] as the backbone for our vision model. To adapt it for our task, we replaced the final layer of the ResNet-50 with two fully connected layers, incorporating ReLU activation, and applied dropout with a probability of 0.3 to mitigate overfitting. Only these replaced layers were retrained, following the transfer learning approach. This strategy significantly reduces training time while mitigating overfitting risks. To create embeddings, we utilized embeddings created by the CLIP model [4].

**Adverse-Weather Dataset** The `Adverse-Weather` dataset comprises numerous RGB image sequences, each with dimensions of $720 \times 1280 \times 3$. These sequences were collected from moving vehicles around the University of Michigan campus, capturing diverse weather conditions. While most sequences exhibit dynamic scenes, some of them include static recordings. The dataset includes two metadata classes: weather and time of day. The weather class consists of labels such as rain, fog, snow, sleet, overcast, sunny, and cloudy. The time of day class includes labels for Sunset, Afternoon, and Dusk. By combining these weather and time of day labels, we established a total of 11 classes to train our model on. To avoid redundancy, we subsampled the images from the video sequences, selecting one image every 10 frames. Consequently, we constructed a dataset comprising 36,230 images, which we divided into a training set of 31,701 images and a test set of 4,529 images.

*Simulation Parameters:* In `Adverse-Weather` simulations, we used 6 heterogeneous environments, each containing 4 robots that observe identical samples, resulting in a total system of 24 robots. To create heterogenous incoming class distributions, we set the skewness parameter of the Dirichlet distribution to $\alpha = 1.2$. In each round, robots are observing 1000 data samples and collect $N^{\text{cache}} = 1$ data samples from observations. We started with an initial dataset of size 10 and collected the data for 10 rounds. The final training dataset consists of 250 data points.

*DNN and Embedding Function:* We adopted a pre-trained Vit-H14 model [67] as the backbone for our vision model. To adapt it to our specific task, we replaced the final layer of the Vit-H14 with two fully connected layers, employing ReLU activation, and incorporated dropout with a probability of 0.3 to mitigate overfitting. Only these replaced layers were subjected to retraining. This methodology significantly reduces training time while effectively preventing overfitting. To create embeddings, we have utilized BADGE [33].

**DeepDrive Dataset:** The `DeepDrive` data following the transfer learning approach, only encompasses 100,000 images taken from driving videos in diverse cities and weather conditions. It comprises 70,000 training images, 10,000 validation images, and 20,000 testing images. However, the testing images are not publicly accessible, so our analysis focused solely on the original training and validation datasets. The classification model was designed to predict weather labels, including rainy, snowy, clear, overcast, partly cloudy, and foggy. However, due to a limited number of foggy images (only 181), the foggy class was excluded from the simulations. Consequently, the classification model was trained on a subset of five classes.

*Simulation Parameters:* In `DeepDrive` simulations, we used 5 heterogeneous environments, each containing 6 robots that observe identical samples, resulting in a total system of 30 robots. To create heterogenous incoming class distributions, we set the skewness parameter of the Dirichlet distribution to $\alpha = 3.3$. In each round, robots observe 1000 data samples and collect $N^{\text{cache}} = 20$

data samples from observations. We started with an initial dataset of size 50 and collected the data for 20 rounds. The final training dataset consists of 12,050 data points.

***DNN and Embedding Function:*** We used a pre-trained Vit-H14[67] model and replaced the final layer of the vision model with 2 fully connected layers with the ReLU activation layer and applied dropout with a probability of 0.3. We only retrained the replaced layers of the network. This approach, known as transfer learning, reduces training time and helps to prevent overfitting. To create embeddings, we utilized BADGE [33].

### A.3.3 Object Detection Experiments

We ran object detection experiments on the DeepDrive dataset. As in the classification case, we used the training and validation images to train our model and report the final mean average precision (mAP) scores. We used an SGD optimizer with a learning rate of 0.01 in this experiment. We trained the DNNs at the start and end of the data collection rounds. We applied all data augmentations detailed in [69].

***Simulation Parameters:*** In object detection experiments, we used 5 heterogeneous environments, each containing 5 robots that observe identical samples, resulting in a total system of 25 robots. We set skewness parameter $\alpha = 1$. In each round, robots observe 1000 data samples and collect $N^{\text{cache}} = 50$ data samples from observations. We started with an initial dataset of size 5000 and collected the data for 20 rounds. The final training dataset consists of 30,000 data points.

***DNN and Embedding Function:*** We used a pretrained YOLOv8-small model [69] and retrained all model weights in each training. Since we are dealing with the objects, we utilized the CLIP model [4] to create embeddings.

## A.4 Object Detection Experiments Metrics

In the object detection experiments, we used several metrics to evaluate the performance of the policies. These metrics include mAP50, representing the mean of average precision at the intersection over union with a threshold of $50\%$; mAP50-95, denoting the mean of the average precision at the intersection over union for thresholds ranging from $50\%$ to $95\%$; precision, and recall. We conducted the experiments with 25 different seeds and averaged the metrics across the seeds. We present results in Table 2. Our INTERACTIVE policy shows a similar performance to the CENTRALIZED policy while outperforming the DISTRIBUTED policy. In mAP50 results, the INTERACTIVE policy shows an improvement of $9.2\%$, whereas, in mAP50-95 results, the INTERACTIVE policy demonstrates a development of $6.1\%$. Furthermore, our INTERACTIVE policy surpasses the DISTRIBUTED policy in precision and recall results by $11.6\%$ and $6.5\%$, respectively.

| Method | mAP@50 | mAP@50-95 | Precision | Recall |
|---|---|---|---|---|
| INITIAL | $33.2 \pm 0.012$ | $17.7 \pm 0.007$ | $51.5 \pm 0.048$ | $32.7 \pm 0.01$ |
| DISTRIBUTED | $37.2 \pm 0.009$ | $20.2 \pm 0.006$ | $56.3 \pm 0.046$ | $35.5 \pm 0.008$ |
| CENTRALIZED | $46.5 \pm 0.003$ | $26.3 \pm 0.007$ | $67.6 \pm 0.007$ | $42.4 \pm 0.004$ |
| INTERACTIVE | $46.4 \pm 0.002$ | $26.3 \pm 0.002$ | $67.9 \pm 0.007$ | $42.0 \pm 0.003$ |

Table 2: Additional metrics for object detection experiments

## A.5 Comparison of Submodular Objective Function Values among Policies

In addition to the accuracy values presented in Fig. 3, we show the values of the submodular objective function across different data collection rounds for the same experiments in Fig. 5. Consistent with the trends observed in accuracy values, our proposed INTERACTIVE policy outperforms the DISTRIBUTED policy and achieves similar performance to the CENTRALIZED policy on all four datasets.

The performance gains of $29.8\%, 19.7\%, 41.2\%,$ and $48.5\%$ for each dataset are achieved because, in the INTERACTIVE policy, a robot makes selections while considering all other robots before it, effectively reducing redundancy., as shown in Fig. 4. These empirical results align with the theoretical analysis presented for the lower-bound in Theorem 1.

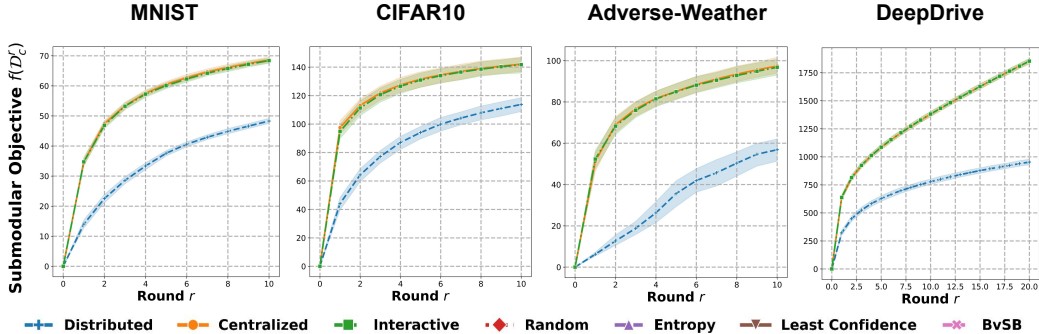

Figure 5: **Evaluating the submodular objective performance of INTERACTIVE, DISTRIBUTED, and CENTRALIZED policies.** This figure illustrates the submodular objective values for the cloud dataset across multiple rounds. Notably, both the CENTRALIZED and INTERACTIVE policies achieve similar objective values, while the DISTRIBUTED policy fails to reach the same level of performance.

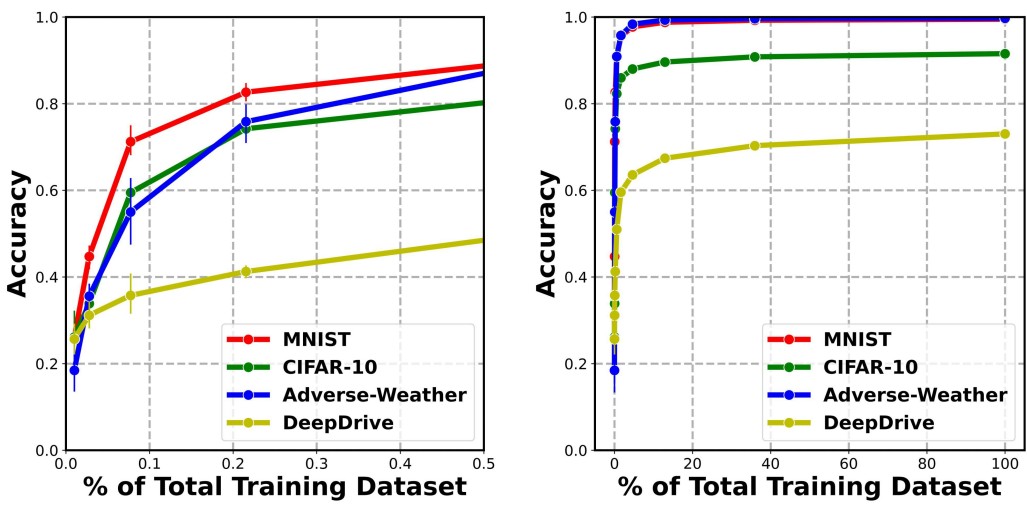

Figure 6: **Accuracy shows diminishing returns to training dataset size.** Each line represents the accuracy of the classification models on a different dataset. The x-axis represents the percentage of datasets used for training the model, while the y-axis represents accuracy. The left figure provides a closer view of the right figure, specifically focusing on training percentages ranging from 0 to 0.5. Across all datasets, we observe a consistent improvement in accuracy as the percentage of training data increases. However, as the dataset size grows, the slope of the curve gradually decreases, indicating diminishing returns where the incremental contributions of new data samples become smaller. This observation supports our assumption of dataset quality function being submodular and monotone.

## A.6 Diminishing Returns between Accuracy and the Percentage of Training Data

To support our claim about dataset quality being submodular in Assumption 3, and monotone 2. We conducted 5 experiments training classification models on four datasets of varying sizes. In Fig. 6, we show the accuracy of the classification models accuracy across the different percentages of the datasets. We can see in the general plot (Fig. 6-right) and zoomed-in version to smaller percentages (Fig. 6-left) accuracy shows a diminishing returns property in terms of accuracy as the training dataset size increases. This indicates that the incremental improvement gained from adding new samples decreases with a larger dataset. Additionally, the accuracy of the models consistently increases as the dataset size grows, providing evidence for the monotonicity of dataset quality. These observations align with our assumption about dataset quality being submodular (Assmp. 3) and monotone (Assmp. 2).

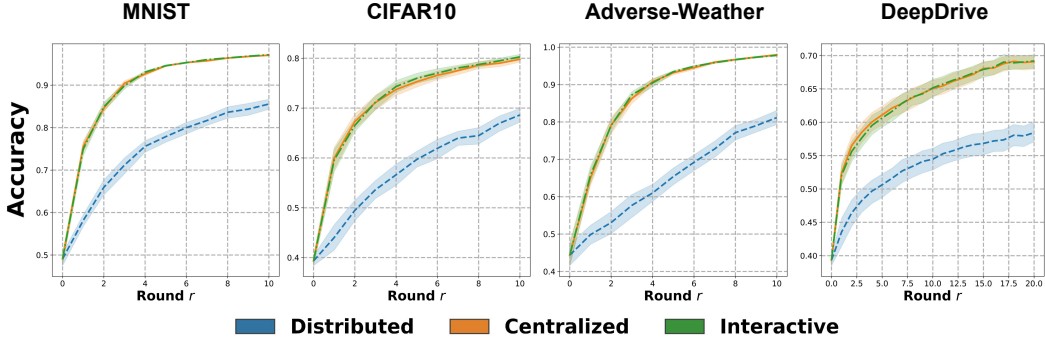

Figure 7: **Comparison of performance under non-monotone and non-submodular objective.** We conducted the same experiments as in Fig. 3 but with a non-submodular and non-monotone objective function, described in A.7. Similar to Fig. 3, we observe that both the INTERACTIVE and CENTRALIZED policies outperform the DISTRIBUTED policy by a significant margin. This demonstrates the robustness and effectiveness of the INTERACTIVE and CENTRALIZED policies under non-submodular and non-monotone objectives.

## A.7   Accuracy When Objective Function is Non-submodular, Non-monotone

In this section, we investigate the performance of our INTERACTIVE policy when the objective function $f$ is not monotone or submodular through experiments. Instead of the submodular facility location function, we employed a variant called the maximin facility location function. This function aims to maximize the minimum distance between any two points in a set. Formally, we can define the maximin facility location function as follows:

$$f_{\mathrm{maximin}}(S) = \max_{s_i \in S} \ \min_{s_j \in S \setminus \{s_i\}} \|s_i - s_j\|$$

where $s_i$ is the $i^{th}$ point in the set $S$ and $\|s_i - s_j\|$ is a distance function. We used the same simulation parameters and models as we did with the facility location objective function. The accuracy results of the simulations are presented in Fig. 7. Similar to the previous case, our INTERACTIVE policy exhibits equivalent performance to the CENTRALIZED policy and outperforms the DISTRIBUTED policy in all simulations.

## A.8   Complexity Analysis

For all algorithms, suppose that there are $N_{\mathrm{robot}}$ robots, each robot $i$ observes samples $X_i^r$ in each data collection round, and in each round at most $N_i^{\mathrm{cache}}$ samples have to be chosen from samples $X_i^r$ for each robot $i$.

It is a known result that the total number of function evaluations of the greedy algorithm that is used for submodular maximization is $O(|V|k)$, where $V$ is the ground set for observations and $k$ is the number of data points that need to be selected [54].

***The* CENTRALIZED *Algorithm:*** In the CENTRALIZED algorithm, a central server carries out all the computation required in a data collection round to choose the samples that are going to be added to the cloud dataset. That is, the central server has access to all the observations made by the robots and has to select $N_i^{\mathrm{cache}}$ samples from the observation set $X_i^r$ of each robot $i$. Therefore, our ground set is $\cup_{i=1}^{N_{\mathrm{robot}}} X_i^r$, and the number of points that need to be selected is $\sum_{i=1}^{N_{\mathrm{robot}}} N_i^{\mathrm{cache}}$. Because the observed sets $X_i^r$ of the robots are disjoint, the total number of elements in our ground set is $|\cup_{i=1}^{N_{\mathrm{robot}}} X_i^r| = \sum_{i=1}^{N_{\mathrm{robot}}} |X_i^r|$. This makes the computational complexity of the CENTRALIZED algorithm in terms of total number of function evaluations $O(\sum_{i=1}^{N_{\mathrm{robot}}} |X_i^r| \times \sum_{i=1}^{N_{\mathrm{robot}}} N_i^{\mathrm{cache}})$.

***The* DISTRIBUTED *Algorithm:*** In the DISTRIBUTED algorithm, all robots carry out the data selection process themselves. Therefore, for each robot $i$, the ground set is $X_i^r$ and the number of points that have to be selected is $N_i^{\mathrm{cache}}$. Summing the number of function evaluations over all robots, the total complexity of the DISTRIBUTED algorithm is $O(\sum_{i=1}^{N_{\mathrm{robot}}} |X_i^r| \times N_i^{\mathrm{cache}})$.

*The* INTERACTIVE *Algorithm:* In our INTERACTIVE algorithm, just like the DISTRIBUTED algorithm, the robots select data points in the confines of their observation sets. Thus, the ground sets and the number of points to be selected from each ground set is again $X_i^r$ and $N_i^{\text{cache}}$, respectively. Summing again over all robots, the total complexity of our INTERACTIVE algorithm is $O(\sum_{i=1}^{N_{\text{robot}}} |X_i^r| \times N_i^{\text{cache}})$, which is the same as the DISTRIBUTED algorithm.

**Number of Message Exchanges:** Since our problem is distributed in nature, we should also analyze the total number of messages passed between robots. The CENTRALIZED algorithm requires $\sum_{i=1}^{N_{\text{robot}}} N_i^{\text{cache}}$ iterations over all robots, resulting in total of $O(N_{\text{robot}} \sum_{i=1}^{N_{\text{robot}}} N_i^{\text{cache}})$ message exchanges. On the other hand, our INTERACTIVE policy requires only one iteration over all robots, leading to a significantly lower number of message exchanges, specifically $O(N_{\text{robot}})$. Lastly, since the DISTRIBUTED policy is executed without any interaction between robots, there are no message exchanges involved.

| Method | Number of Message Exchanges | Number of Function Evaluations |
|---|---|---|
| DISTRIBUTED | - | $O(\sum_{i=1}^{N_{\text{robot}}} |X_i^r| \times N_i^{\text{cache}})$ |
| CENTRALIZED | $O(N_{\text{robot}} \sum_{i=1}^{N_{\text{robot}}} N_i^{\text{cache}})$ | $O(\sum_{i=1}^{N_{\text{robot}}} |X_i^r| \times \sum_{i=1}^{N_{\text{robot}}} N_i^{\text{cache}})$ |
| INTERACTIVE | $O(N_{\text{robot}})$ | $O(\sum_{i=1}^{N_{\text{robot}}} |X_i^r| \times N_i^{\text{cache}})$ |

Table 3: Computational Complexities of Different Policies

### A.9 Numerical Evaluation of Policy Performance Metrics

To provide a quantitative example of the analysis conducted in Section A.8, we offer numerical instances from experiments carried out on the `DeepDrive` dataset, detailed in Table 4. As evidenced in the table, there are noteworthy distinctions in terms of computation and communication metrics among the compared policies.

Specifically, in terms of message exchanges, our proposed INTERACTIVE policy showcases exceptional efficiency, outperforming the CENTRALIZED policy by a factor of $600\times$. Moreover, in relation to function evaluations, both the INTERACTIVE and DISTRIBUTED policies demonstrate efficiency improvements of $20\times$ compared to the CENTRALIZED policy.

With regards to accuracy, an improvement of $9.4\%$ is achieved by both the CENTRALIZED and INTERACTIVE policies. This signifies that our proposed INTERACTIVE policy attains accuracy levels akin to those of the CENTRALIZED policy, with the addition of only $30$ additional message exchanges among robots, compared to the DISTRIBUTED policy. This, in turn, positions our proposed policy as significantly more scalable than the CENTRALIZED policy.

| Method | No. of Message Exchanges | No. of Function Evaluations | Accuracy |
|---|---|---|---|
| DISTRIBUTED | - | 600,000 | 0.65 |
| CENTRALIZED | 18,000 | 12,000,000 | 0.75 |
| INTERACTIVE | 30 | 600,000 | 0.75 |

Table 4: Performance Metrics for *DeepDrive* experiments

### A.10 Federated Learning Experiments and Comparative Analysis

Federated learning (FL) is a commonly used method to train machine learning models in decentralized settings [73]. In FL, models are trained in a distributed fashion, where individual robots train models locally and transmit only gradient updates to a central cloud. These gradient updates are then aggregated, producing an averaged gradient that, in turn, updates the global model. FL preserves the privacy of local data, as only gradient updates are shared with the cloud, followed by their averaging.

To compare our method against FL methods, we trained FL models using the local data from each robot. We simulated scenarios where each robot had labels for randomly selected subsets of local observations, corresponding to $1\times, 2\times, 5\times, 10\times, 20\times$, and $50\times$ the cache size in the active learning counterpart. In all experiments, each robot independently trained its local model for 20 epochs using the same optimization parameters outlined in Section A.3. Subsequently, each robot shared

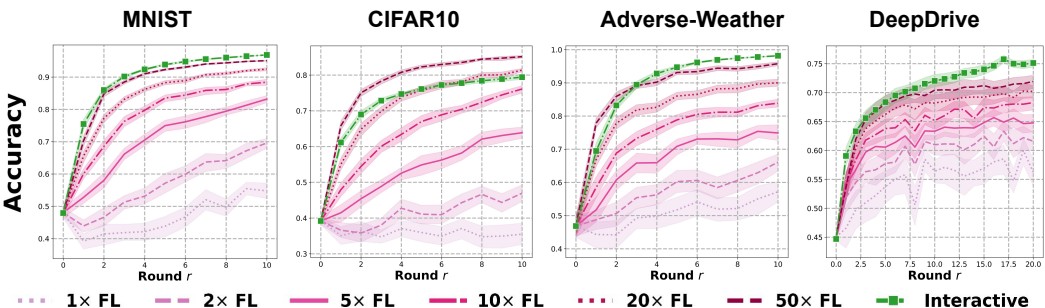

Figure 8: **Comparison of our INTERACTIVE policy versus Federated Learning.** In addition to the active learning benchmarks, we have compared our INTERACTIVE policy against a federated learning scheme. The different shades of purple illustrate the FL experiments, where the local training of the ML model employs $1\times, 2\times, 5\times, 10\times, 20\times$, and $50\times$, the amount of data used in our active learning scheme. Darker shades indicate larger data amounts in the FL experiments. Across all experiments, the results consistently demonstrate the superior performance of our proposed INTERACTIVE policy when the same number of data points is used in FL. Remarkably, the FL approach requires up to $50\times, 20\times$, and $50\times$ more data to attain performance comparable to our policy on the MNIST, CIFAR-10, and Adverse-Weather datasets, respectively. This underscores the remarkable data efficiency of our approach.

gradients with the cloud. At the end of each round, the cloud averaged these gradients to update the global model. This updated global model was then sent back to each robot in the network.

The results for the FL experiments and our proposed policy are presented in Fig. 8. Across all the experiments, when our proposed policy and FL have access to the same amount of data, our policy consistently outperforms the FL counterparts by $42.1\%, 43.9\%, 41.0\%$, and $18.9\%$. In addition, to achieve the same accuracy as our policy, FL needs $50\times, 20\times$, and $50\times$ more labeled data samples for the MNIST, CIFAR-10, and Adverse-Weather datasets, respectively. However, in the more challenging DeepDrive dataset, FL is not able to achieve a comparable result even when $50\times$ more labeled data samples - all of the local observations - are used in the training. This observation further underlines the efficiency of our active learning-based approach relative to FL.

Several factors contribute to this performance divergence. In the context of active learning, the selection of data points for each robot is limited to a maximum of 20 across all simulations. As a result, when FL is trained on the same number of samples, the local models tend to overfit, thereby compromising the performance of the global model. Moreover, for more challenging ML tasks, the local dataset size within FL falls short of achieving high accuracy, even when all observed data points are utilized. In contrast, our proposed INTERACTIVE policy mitigates these issues by pooling the data in a centralized cloud, enhancing the training dataset size of the global model. As rounds progress, the shared data pool expands, leading to improved overall model performance. In this light, our INTERACTIVE policy complements traditional FL methods.

