# OpenReview forum: "Fleet Active Learning: A Submodular Maximization Approach"
_robot-learning.org/CoRL/2023/Conference — CoRL 2023 Poster_

### Official Review · Reviewer_25hL · 2023-07-13

**Confidence:** 2
**Originality:** Very Good
**Technical Quality:** Good
**Clarity Of Presentation:** Very Good
**Impact:** 3

**Recommendation:**

Weak Accept: I recommend accepting the paper, but will not argue for my recommendation if the majority of other reviewers have a different opinion.

**Review:**

Review:

Strengths:
- Nice overview of related work, especially explicitly mentioning the differences between the proposed method to related works is very valuable
- Nice writing. The paper is easy to read such that the reader can easily follow the story.
- Promising results


Weaknesses:
- No major weaknesses, however, additional comparisons would be nice (see Questions for Rebuttal)

**Quality Of The Limitations Section:**

Additional details required

**Questions For Rebuttal:**

- Were experiments with different similarity measures conducted?
- How is the data labeling performed?
- It would be interesting to see how federated learning (FL) performs if labeled data was provided. It might be that it outperforms the proposed method but a comparison would still be very interesting.




**Robotics Focus:**

Highly relevant to robotics but no hardware experiments

**Summary Of Paper:**

This paper tackles the problem of intelligently selecting useful data collected by a multi-robot system under constraints. Each robot collects data, but only some of them are useful as other robots might have collected similar data already. The collected data is sent to a cloud server, where the underlying machine learning model is updated.
The paper proposes to allow communication between the robots to effectively select a subset of their newly collected data sets based on a submodular objective that involves similarity measures between data points. The whole data set can not be sent to the cloud due to bandwidth constraints.
The proposed method is shown to outperform the baseline and achieve the same performance as a centralized method which assumes to have access to the whole data set.

**Summary Of Recommendation:**

This paper presents an interesting approach to Fleet robotics with promising results. I believe it is relevant to the research community.

---

> ### Author Response · Authors · 2023-08-14
> **New Federated Learning Experiments**
>
> We have revisited your request for **comparisons with a federated learning** (FL) setup and conducted experiments that compare our proposed active learning (AL) policy with an FL approach. As outlined in our main paper, it's important to note that FL is computationally intensive, particularly for large models. Consequently, these experiments required 3 days to complete and incurred significant computational costs on our cluster.
>
> Our recently conducted experiments confirm that our approach **outperforms FL in the data-limited regime**, where FL and our policy use similar amounts of data. In experimental scenarios where the task is relatively easy, as the number of data points used in FL increases, it achieves similar or even superior performance. However, for more challenging datasets, FL falls short of outperforming our method. For instance, in the MNIST experiments, achieving the equivalent accuracy of our algorithm requires up to $50 \times$ more labeled data samples in the FL approach. Conversely, in the DeepDrive experiments, FL fails to outperform our policy even when all data points are utilized.
>
> Several factors contribute to this performance gap. In our active-learning-based policy, each robot utilizes a maximum of 20 data points in each round. Consequently, when FL employs the same number of points, it can lead to local model overfitting, thereby decreasing the overall global model performance. Furthermore, in more challenging ML tasks, the local dataset size proves inadequate for achieving high accuracy, even when all observed data points are utilized. In contrast, our Interactive policy accumulates data in a centralized cloud, expanding the global model's training dataset. The shared data pool grows as the rounds progress, consistently improving model performance and resulting in higher accuracy compared to the FL counterpart.
>
> We have included new results and a more detailed analysis in **Appendix A.9**.
>
> We believe these new experiments, with the addition of new active learning benchmarks, show the effectiveness of our method and significantly increase the strength of our work.
>
> We would be happy to answer additional questions before the Rebuttal period ends.

---

### Official Review · Reviewer_fsBm · 2023-07-19

**Confidence:** 3
**Originality:** Good
**Technical Quality:** Good
**Clarity Of Presentation:** Very Good
**Impact:** 3

**Recommendation:**

Weak Accept: I recommend accepting the paper, but will not argue for my recommendation if the majority of other reviewers have a different opinion.

**Review:**

Overall, the paper is well written, easy to follow and studies a relevant and active problem. The proposed approach is technically sound and comes with a thorough theoretical analysis. The simulations are using real-world data sets with a variety of different configurations and showcase advantages of the proposed method. However, I have a few comments that should be clarified, please see below.



**Quality Of The Limitations Section:**

Additional details required

**Questions For Rebuttal:**

1.	Assumptions 2 and 3: I understand the intuition behind these assumptions; however I think that they need to be discussed more thoroughly. Any ML system can run into overfitting. Thus, couln't adding more samples become detrimental to the dataset quality if the new data only represents a small part of the distribution? For instance, if we ‘spam’ the dataset with a large amount of data recorded under certain road conditions, I would suppose that this can lead to poorer performance for other conditions.
Similarly, the diminishing return might not hold: Suppose we already have some data, but it does not cover, a certain scenario yet. The first datapoint for that scenario might not be of much value since it is insufficient to train any new behaviour. Yet, adding additional data from that scenario might eventually lead to a better network output, which would invalidate submodularity.
It is hinted that “the dataset already covers a significant portion of the real data distribution” – this needs more elaboration to justify assumptions 2 and 3 (Also the simulation setup does not support this since the initial datasets are quite small). Lastly, the appendix includes an empirical study that the accuracy measure is showing a diminishing return – while this provides additional motivation for the approach, it does not substitute a more rigorous theoretical justification.

2.	NP hardness: The paper simply states that the problem is a combinatorical optimization problem and is NP hard. There are many combinatorial problems that are not NP-hard. Hence, this needs some more justification, e.g., a reduction (I realize that space is limited) or clearly relating it to a specific NP-hard problem, or giving a citation that considered this problem and showed hardness.

3.	Simulation results: It is surprising that there is no gap between the proposed iterative approach and centralized. The problem statement states that each robot has X_i^r datapoints available – yet from the experiment description and the appendix it is not clear if these are the same sets for all robots or if they differ. If they are the same, it is clear that iterative would perform the same as centralized, but this might not be the most interesting problem setup to consider. Furthermore, in 2 experiments, each robot selects only 1 sample per round, in another it is 3 samples; only for DeepDrive robots select 20 samples. I think that the small number of selected samples might also lead to the two algorithms being quite similar.

4.	Simulations: It would be good if the experiments could highlight the shortcomings of the centralized approach. While the bandwidth constraints and computational burden are mentioned in the introduction, the current results do not show any disadvantage of centralized and rather focus on showing that the proposed method is equally good w.r.t. the submodular objective and the accuracy. I noticed the complexity analysis in the appendix, but some numerical evaluation would bring additional practical insights.

Minor comments:

5.	Fig 1: Could the submodular function f be somehow included in the figure?

6.	Definition 1: I think it is a bit unusual to use a lower case for a set (here a_i^r).

7.	Structure: Why are the baselines (Algorithms 1 and 2) part of the problem formulation? I think these should be in a new section. Also after Assumption 3, the main problem could be put in a theorem-style environment to make it stand out more.

8.	Line 253: Theorem 1 has shown that the proposed policy has the same worst case bound as the centralized one, but that does not imply that it would perform as good as centralized.

**Robotics Focus:**

Highly relevant to robotics but no hardware experiments

**Summary Of Paper:**

The paper studies how autonomous vehicles (AV) can improve their datasets for perception and planning by actively selecting new datapoints for training. When deploying a fleet of AVs it is impractical to collect and process all available data due to bandwidth constraints and computational cost. Active learning methods allow for choosing only the most informative datapoints, yet distributed execution on multiple vehicles can lead to redundant data and thus poor performance. The authors propose a hybrid method where all AVs iteratively select their next data set. Here, each vehicle builds on the data collected by all AVs in previous rounds. The problem is formulated as a submodular maximation, and the authors show that their approach is a 2-factor approximation. Simulation results suggest that the method performs equally well as a centralized approach and clearly outperforms a purely distributed approach where robot exchange no information.


**Summary Of Recommendation:**

Overall, this is a well-written and original paper with a solid contribution. However, my two main concerns are the assumptions about monotonicity and submodularity that need further discussion, and some clarification on the simulation results since it is quite surprising that there is no gap between centralized and the proposed.

---

> ### Author Response · Authors · 2023-08-14
> **New Federated Learning Experiments**
>
> We have introduced new experimental findings that directly compare our active learning policy with its federated learning (FL) counterpart. We believe these additional experiments significantly strengthen our paper, as they demonstrate that our method is considerably more data-efficient than its FL counterpart. We have included the new results and a more detailed analysis in **Appendix A.9**.
>
> We welcome any inquiries you may have before the conclusion of the Rebuttal period.

---

### Official Review · Reviewer_xJp2 · 2023-07-20

**Confidence:** 4
**Originality:** Good
**Technical Quality:** Good
**Clarity Of Presentation:** Good
**Impact:** 2

**Recommendation:**

Weak Accept: I recommend accepting the paper, but will not argue for my recommendation if the majority of other reviewers have a different opinion.

**Review:**

Pros:
- the paper grounds itself in practical scenario of data gathering for autonomous cars. Perhaps in control literature, there has been similar set ups with distributed systems/sensing, e.g., sensor network placement, etc, the paper brings these ideas for practical robotic scenarios.
- the paper is generally well written in terms of explanations. Especially, the paper clearly outlines the assumptions and settings.


Cons:
- in the experiments, the choice of baselines should be motivated.
The paper chooses a completely distributed approach and a centralized approach as baselines. Why not add a random baselines, which picks from the complete data pool (like centralized) and the data pool of each robots (like decentralized)? While experiments do validate the design of the approach to certain degrees, readers may appreciate more, when a thorough comparison to basic baselines (random uniform sampling) and other pool-based active learning methods for distributed systems, e.g, uncertainty sampling.

- the claimed contributions and related work require clarifications.
The paper states that one of its contribution is to frame the data collection problem of multiple robots as submodular maximization problem. Does not reference 46 and 47 also use submodular maximization approach for similar problem, where the goal is to maximize the information gain? It would help to comprehend the contributions of the paper.

I also think that some of the statements in the abstract can be misleading, The statements on the performance in particular, is with respect to one baseline, which is completely decentralized, while the paper utilizes certain communication with some of the robots. I am not sure, if that would be a fair comparison, and I would appreciate for the clarifications.

**Quality Of The Limitations Section:**

Limitations are addressed clearly

**Questions For Rebuttal:**

See the section above.

**Robotics Focus:**

Highly relevant to robotics but no hardware experiments

**Summary Of Paper:**

This paper presents an active learning framework for a distributed learning system, i.e., a fleet of autonomous vehicles, collecting data for a perception system on the roads. Ideally, the data collected is limited in size, but at the same time, the most informative and diverse for a neural network to learn from. The paper builds upon the previous work of submodularity for active learning, and provide a solution called INTERACTIVE, which minimizes communication between the agents, while distributed maximizing the information gain. Experimental results show that the proposed solution yields similar results to the completely centralized approach, while outperforming the purely distributed one.

**Summary Of Recommendation:**

I generally find the paper interesting and relevant for many problems in robotics. The paper is also well written, and has rigours for the presented method. I however would like to see some of the clarifications to the raised issues above, before making the final recommendations.

------------------------------------ Post rebuttal ----------------------------------

I have increased the score to weak accept, since the authors have addressed most of my comments through experiments.

---

> ### Author Response · Authors · 2023-08-14
> **New Federated Learning Experiments**
>
> In addition to the extra active learning benchmarks, we conducted experiments to compare our method with a federated learning (FL) setup. These new experiments took us three days and required significant computational power on our cluster. These additional results are also interesting as they demonstrate the data efficiency of our method relative to the FL in the data-limited regime. Notably, for MNIST, FL needs up to 50 times more labeled data to achieve equivalent accuracy. However, in the more challenging DeepDrive dataset, FL cannot surpass our policy even when all local data points are labeled and utilized.
>
> Two main factors contribute to this performance difference. First, our proposed policy only utilizes up to 20 data points across all experiments. When a similar number of data points is used for FL in local training, local models tend to overfit, leading to decreased global model accuracy. Second, the number of local observations is insufficient for more demanding ML tasks. In contrast, our active-learning-based Interactive policy accumulates data in a centralized cloud, expanding the global model's training dataset. The shared data pool grows as the rounds progress, consistently improving model performance and resulting in higher accuracy than the FL counterpart.
>
> We have included the new results and more detailed analysis in **Appendix A.9**. We believe that these new experiments, along with the new active learning benchmarks, significantly increase the comprehensiveness of our work.
>
> We would be happy to answer additional questions before the Rebuttal period ends.

---

### Author Response · Authors · 2023-08-09
**Revised Paper and Appendix are Attached to Rebuttals**

Dear Reviewers,

Thank you for your valuable feedback. We present our revised paper and appendix, incorporating all reviewers' comments.

Highlights of the Revision:

1. Figure 3 is updated with additional results from classical active learning benchmarks. These additional results significantly bolster the comprehensiveness of our work.
2. Table 3 is introduced to subsection A.7., which concisely compares computational complexities across various policies.
3. A new subsection named “Numerical Evaluation of Policy Performance Metrics” is added to the appendix. This section presents numerical values for the number of message exchanges, function evaluations, and model accuracy in the DeepDrive dataset experiment.

Revisions are highlighted in blue for your convenience.

Thanks,

The Authors

---

### Comment · Area_Chair_r41Q · 2023-08-11
**Discussion until Aug 15th**

I would like to thank the authors and reviewers and would like to encourage you to make best use of the discussion period which will end *Aug 15 at 11:59 PM PT*.

In particular, to the reviewers: you are highly encouraged to engage in additional discussions with the authors if there are any remaining questions you would like the authors to elaborate on further.

---

### Decision · Program_Chairs · 2023-08-30

**Decision:**

Accept (Poster)

**Comment:**

The authors study a fleet active learning approach where autonomous agents can select training data samples to improve local models. The chosen approach in particular utilizes submodular maximization in the sample selection process. The review process resulted in a uniform recommendation of "weak accept" among the reviewers. Among the strengths of the work highlighted by the reviewers were
in particular that the work is well written and readable, that the grounding of the problem was well motivated and potentially relevant to many problems in robotics. The questions raised by one reviewer regarding the submodulariy assumptions are I believe an area that may warrant additional future investigation in particular and a careful discussion of potential edge or failure-cases could further strengthen the work.